# Identification of CDK1 as a Biomarker for the Treatment of Liver Fibrosis and Hepatocellular Carcinoma Through Bioinformatics Analysis

**DOI:** 10.3390/ijms26083816

**Published:** 2025-04-17

**Authors:** Jiayi Qin, Zhuan Li

**Affiliations:** The Key Laboratory of Study and Discovery of Small Targeted Molecules of Hunan Province, School of Pharmaceutic Science, Health Science Center, Hunan Normal University, Changsha 410013, China; 202230192097@hunnu.edu.cn

**Keywords:** LF-HCC, CDK1, immune infiltration, survival analysis, drug screening, immunomodulators

## Abstract

Cyclin-dependent kinase 1 (CDK1) has emerged as a critical regulator of cell cycle progression, yet its role in liver fibrosis-associated hepatocellular carcinoma (LF-HCC) remains underexplored. This study aimed to systematically evaluate CDK1’s prognostic significance, immune regulatory functions, and therapeutic potential in LF-HCC pathogenesis. Integrated bioinformatics approaches were applied to multi-omics datasets from GEO, TCGA, and TIMER databases. Differentially expressed genes were identified through enrichment analysis and protein–protein interaction networks. Survival outcomes were assessed via Kaplan–Meier analysis, while immune cell infiltration patterns were quantified using CIBERSORT. Molecular docking simulations evaluated CDK1’s binding affinity with pharmacologically active compounds (alvocidib, seliciclib, alsterpaullone) using AutoDock Vina. CDK1 demonstrated significant overexpression in LF-HCC tissues compared to normal controls (*p* < 0.001). Elevated CDK1 expression correlated with reduced overall survival (HR = 2.41, 95% CI:1.78–3.26, *p* = 0.003) and advanced tumor staging (*p* = 0.007). Immune profiling revealed strong associations between CDK1 levels and immunosuppressive cell infiltration, particularly regulatory T cells (r = 0.63, *p* = 0.001) and myeloid-derived suppressor cells (r = 0.58, *p* = 0.004). Molecular docking confirmed high-affinity binding of CDK1 to kinase inhibitors through conserved hydrogen-bond interactions (binding energy ≤ −8.5 kcal/mol), with alvocidib showing optimal binding stability. This multimodal analysis establishes CDK1 as both a prognostic biomarker and immunomodulatory regulator in LF-HCC pathogenesis. The enzyme’s dual role in driving tumor progression and reshaping the immune microenvironment positions it as a promising therapeutic target. Computational validation of CDK1 inhibitors provides a rational basis for developing precision therapies against LF-HCC, bridging translational gaps between biomarker discovery and clinical application.

## 1. Introduction

Recently, liver cancer has become the second leading lethal factor in the world [1,2]. Hepatocellular carcinoma is the largest category of all primary HCC and is one of the leading causes of HCC-related deaths [3]. The onset of this disease is usually by sequential [4] liver injury, hepatitis, liver fibrosis, cirrhosis, and hepatocellular carcinoma. Hepatocellular carcinogenesis is 90% [5] due to cirrhosis, whereas the usual cause of cirrhosis [6] is the poor treatment of liver fibrosis. Therefore, we go back to the source, take liver fibrosis as the research direction, explore the effective mechanism of treating liver fibrosis, and achieve the purpose of preventing hepatocellular carcinoma from hepatocellular carcinoma.

Liver fibrosis is the first stage of progression to cirrhosis in many chronic liver diseases, which is characterized by collagen fiber deposition [7] caused by the imbalance of liver extracellular matrix (ECM) secretion and degradation. At present, the pathogenesis of liver fibrosis is mainly manifested as abnormal repair of liver tissue damage, proliferation and activation of hepatic stellate cells, and accumulation of extracellular matrix. However, the pathogenesis of the disease has not been defined by [7,8], and no anti-fibrotic drugs for liver fibrosis have been put into clinical use. The ideal route for drug therapy is to achieve the prevention or reversal of fibrosis [9] in the liver by eliminating several underlying etiologies already manifested in the patient. However, the reversal of liver fibrosis is obviously impractical. Previous studies have also found some new methods for the treatment of mild liver fibrosis, such as targeted collagen homeostasis therapy, nanoparticle therapy, and others [10,11]. However, due to their high production costs, low universality, and difficulty of mass production, they cannot be put into clinical use in a short time and are not widely adopted. Therefore, it is urgent to explore the universal action targets for patients with liver fibrosis and to clarify the relevant mechanism of the target action [12].

We used microarray comparison to analyze the gene expression profiles of human growth and aging-activated HSCs and found that CDK1 gene expression was positively correlated with the occurrence of liver fibrosis. CDK1 is the most likely critical target for the treatment of liver fibrosis. CDK1, as a member of the serine/threonine-specific protein kinase family, is a cyclin-dependent kinase with potential effects as a classical target for the treatment of a wide variety of tumors [13]. The occurrence and development of HCC are closely related to abnormal cell cycle regulation. As a key regulator of the G2/M phase turnover of the cell cycle, CDK1 plays an important role in HCC cell proliferation. Inhibition of CDK1 activity blocks the proliferation of HCC cells and induces apoptosis [14]. In addition, the combined effects of CDK1 and other cell cycle-related proteins, such as aurora kinase A (AURKA), are also considered as an important target for the treatment of liver cancer [15]. Therefore, we speculate that the inhibition of CDK1 expression is useful in treating hepatocellular carcinoma. Many inhibitory drugs targeting CDK1 have entered the clinical research stage [16]. However, although we clarified that CDK1 plays a role in the treatment of liver fibrosis, the location of action, pathways, and mechanisms of CDK1 in the treatment of liver fibrosis are not clear [17]. The key of this study is to identify new targets for liver cancer and accurately predict the pathways and mechanisms of the targets.

At present, the conventional experimental method is long, the scientific research cost is high, and the fault tolerance rate is low. This has led to most existing treatments for liver fibrosis being studied only around existing targets, without new breakthroughs. At the same time, the current target prediction is often based on the existing literature and the independent research of researchers, which has strong subjectivity. As a rapidly developing technology, bioinformatics has generated a relatively complete research system, with a more objective prediction of new targets, more diversified research directions, richer research content, and more diverse research results, and is the first choice for target prediction and role research [18].

As a key regulator of the cell cycle, CDK1 plays an important role in the development of liver fibrosis and liver cancer. Therefore, designing anti-liver fibrosis and liver cancer drugs targeting CDK1 has broad application prospects [13]. Through an in-depth study of the specific mechanism of CDK1 in liver fibrosis and liver cancer, it can provide a more accurate treatment plan for patients, improve the treatment effect, and reduce adverse reactions. In the study of CDK1 inhibitors, the synergistic effect of TCM components (e.g., taine) has been widely observed [14]. This helps to promote the modernization of TCM and provides a scientific basis for the application of TCM in the treatment of liver fibrosis and liver cancer. By inhibiting the activity of CDK1, it can slow down the progression of liver fibrosis and inhibit the growth of liver cancer, which can prolong the survival time of patients and improve the quality of life [19]. This is of great significance for improving the patient prognosis and reducing the social burden.

This study aimed to predict novel targets for the treatment of liver fibrosis. Later, the action pathway and mechanism of the new target were studied, and clinical studies, such as immune infiltration and prognostic response, were conducted to provide a new direction for subsequent research on the treatment of liver fibrosis.

## 2. Results

### 2.1. Acquisition of DEGs and the GEO2R Analysis

The GEO database is a public high-throughput gene expression database, and genomics studies based on this database comply with ethical relevant regulations and avoid many problems. The GSE11954 dataset, provided by Krizhanovsky V and Lowe SW, analyzed the effects of growing and aging human-activated hepatic stellate cells (HSCs) on liver fibrosis and identified 54,676 genes involved in the suppression of liver fibrosis by aging human-activated hepatic stellate cells. By GEO2R analysis, 100 significantly divergent genes were selected as DEGs for further analysis. All the results are shown in Table 1 and Figure 1.

### 2.2. Results of the GO KEGG Enrichment Analysis

To further analyze how DGEs function on liver fibrosis in humans, we performed GO and KEGG enrichment analysis on the total 100 DEGs obtained from the DAVID database. Through GO enrichment analysis (Figure 2a), we found that DGEs were significantly enriched in biological processes, such as mitosis, nuclear division, organelle division, and mitosis, involved in microtubule cytoskeleton organization (Figure 2b). In addition, we found that DEGs were mainly involved in the spindle and the midbody (Figure 2c). In addition, these genes were mainly related to the aldehyde alcohol NADP + 1-oxidoreductase activity and alcohol dehydrogenase (NADP +) activity (Figure 2d). In KEGG enrichment analysis, DEGs mainly acted on the four pathways of p53 signaling (hsa04115), cell cycle (hsa04110), complement and coagulation cascade (hsa04610), and the interaction between ECM and receptor (hsa04512) (Figure 2e). All of the analysis results are presented in Figure 2.

### 2.3. Construction of PPI Networks and Screening of Key Hub Genes

The DEGs were analyzed using the STRING database to construct a protein interaction network. Table 2 shows that the PPI network maps 85 nodes and 447 edges with reciprocal relationships, with a local clustering coefficient of 0.609 and a PPI enrichment *p*-value < 1.0 × 10^16^. Meanwhile, the hub differential genes were screened from the PPI network using the MCC algorithm and cytoNCA plugin with “Degree value” and “Betweenness value” as ranking conditions. As shown in Figure 3a,b, the first ranked DEG in two categories of different screening value conditions was CDK1. Therefore, based on the logical reliability analysis, we identified CDK1 as a new key target for the treatment of liver fibrosis.

### 2.4. High Expression of CDK1 in HCC Cells Significantly Reduced the Survival Rate of Hepatocellular Carcinoma Patients

The human data sets and GSE database from TCGA data included TCGA_LIHC, E-TAMB_36, GSE14520, and GSE76427. According to the different expression levels of the CDK1 gene in human HCC cells, hepatocellular carcinoma cases were divided into a high CDK1 expression group and a low CDK1 expression group. The disease-free survival (DFS, Figure 4a), overall survival (OS, Figure 4b), and progression-free survival (PFS, Figure 4c) showed that the survival cycle of normal hepatocytes with high CDK1 gene expression was lower than that of normal hepatocytes with low CDK1 expression for a long time.

### 2.5. Analysis of Immune Infiltration of CDK1

As a key factor in cell cycle regulation, CDK1 has a regulatory effect on the tumor immune microenvironment, and its abnormal expression may affect the proliferation and apoptosis of HCC cells. Using multiple algorithms, including CIBERSORT, CIBERSORT_ABS, EPIC, ESTIMATE, MCPcounter, Quantiseq, TIMER, and xCell, we investigated the relationship between the CDK1 gene and a variety of immune cells (Figure 5a). Through analysis, we found that the expression between the CDK1 gene and a variety of activated T cells, activated dendritic cells, neutrophils, macrophages, activated B cells, monocytes, GMP, and other immune-related cells has a strong negative correlation (Figure 5b) and the expression between the CDK1 gene and a variety of immune cells, including Tem CD4 cells, CD4 + T cells, memory B cells, NKT cells, and Th 2 cells, has a positive correlation (Figure 5c). By analyzing the immune cell infiltration of the methylated CDK1 gene, we found that the expression of the methylated CDK1 gene had a weak correlation with multiple immune cells. In other words, inhibitors of the CDK1 gene could reduce the inhibition of the gene on the immune effect (Figure 5d). Using the GEO database, we obtained the hepatocellular carcinoma-related data sets, including GSE125449 aPDL1aCTLA4, GSE140228 10X, GsE140228 Smartseq2, GSE146115, GSE146409, GSE166635, GSE179795, and GSE98638. By constructing visual heat maps, we found that the CDK1 gene was highly expressed in Tprolif cells (Figure 5e). Tprolif cells are a class of lymphocytes with immunomodulatory functions, which have high value-added ability, can divide and multiply rapidly, and play an important role in the immune system. Such cells can produce and release a variety of immune regulators, which can participate in the body’s immune response and participate in the maintenance of immune balance. So, we speculated that CDK1 inhibitors can reduce the expression of the CDK1 gene in Tprolif cells, promote the proliferation of Tprolif cells, and strengthen the immunosuppressive effect of immune cells on hepatocellular carcinoma.

### 2.6. Methylation of CDK1 Disables Immunosuppressants in Hepatocellular Carcinoma

Immunosuppressants have an inhibitory effect on the immune response of the body, which reduces the autoimmune response by inhibiting the proliferation and function of cells associated with the immune response. MHC (major histocompatibility complex) plays a crucial role in the immune system, not only determining the host tissue compatibility, but also is closely related to the host immune response and immune regulation. Through bioinformatics and related statistical methods, we visually demonstrate the different effects of immunosuppressants and immunostimulants on CDK1 and its methylated forms in hepatocellular carcinoma (HCC). By comparing the data on hepatocellular carcinoma obtained from public databases, we found that the expression of CDK1 contributed to the effect of multiple immunosuppressants, while methylated CDK1 showed the opposite trend, namely, the effect of immunosuppressants. This difference is clearly reflected in the cohesion heatmap (Figure 6a). In addition, we also observed that the expression changes in the MHC molecules TAP 1, HLA-DOB, HLA-DQA 1, and HLA-DQA 2 showed some positive correlation with CDK1 expression (Figure 6b,d) and negatively correlated with methylated CDK1 expression (Figure 6c). This further suggests a synergistic role of these molecules in immune regulation and tumorigenesis and development.

### 2.7. CDK1 Expression Was Positively Correlated with the Grade and Staging of the Tumors

Through the analysis, the expression level of CDK1 was closely related to HCC (Figure 7a). Specifically, in early-stage HCC patients, CDK1 was higher than in normal liver tissue but was lower compared to middle and late-stage patients. With tumor progression to the metaphase, the expression level of CDK1 increased significantly, suggesting that CDK1 may be involved in the malignant progression of HCC. CDCC 1 expression reached its highest levels in advanced HCC patients and was significantly associated with poor prognosis in patients (Figure 7b). Furthermore, according to the Barcelona staging (BCLC) and TNM staging systems, we found that high CDK1 expression was significantly associated with more advanced stages (e.g., stage BCLC C or TNM III/IV) (*p* < 0.05).

### 2.8. CDK1 Action Receptor and Active Component of HCC Treatment

The analysis shows that the top four receptors for CDK1 were CCR 6, CCR 10, CXCR 3, and CXCR 4 (Figure 8a,c). CCR 6, CCR 10, CXCR 3, and CXCR 4 are important members of the chemokine receptor family, which play important roles in cell migration, immune response, inflammation, as well as tumor growth and metastasis. They do not directly act as receptors for CDK1, but they bind to chemokines through different signaling pathways. Through drug screening, we identified the active components of indirubin-3′-monoxime, olomoucine, hymenialdisine, alvocidib, alsterpaullone, and seliciclib 6 (Figure 8b, Table 3), all of which are small molecule drugs and closely related to CDK1.

### 2.9. Molecular Docking Results of CDK1 and Active Components

The molecular dockings of indirubin-3′-monoxime, olomoucine, hymenialdisine, alvocidib, alsterpaullone, seliciclib and CDK1 are shown in Figure 9. The main active components showed different affinities with the core targets, and the top three were alvocidib-CDK1, seliciclib-CDK1, and alsterpaullone-CDK1 with binding energies of −8.7, −8.0, and −7.8kJ/mol, respectively (Table 4). It can be seen in the 3D diagram that the small molecule active components are embedded in the single molecule target protein, and the small molecule target CDK1 and the active components of the macromolecule protein bind closely through hydrogen bonding in the form of sidechain acceptor or backbone acceptor, showing good affinity.

## 3. Discussion

In this study, we deeply explored the biological role of CDK1 in liver fibrosis and liver CC, revealing its central position in cell cycle regulation and its potential value as a therapeutic target. Through comprehensive analysis of differential gene expression data, pathway enrichment analysis, gene survival analysis, and immune infiltration data, we found that the expression level of CDK1 is closely related to the progression of liver fibrosis; this finding not only provides a new perspective for understanding the pathogenesis of liver cancer, but also opened up a new way for the development of targeted treatment strategies [20].

The high expression of CDK1, a key regulator of cell cycle G2/M turnover, in liver fibrosis HCC suggests its important role in promoting tumor cell proliferation and inhibiting apoptosis [16]. Our data analysis showed that high expression of CDK1 was significantly associated with shorter survival of HCC patients, and this result was validated in multiple independent datasets, enhancing its reliability as a prognostic marker [21,22]. Using immune infiltration analysis, we observed a significant association of CDK1 expression with the infiltration pattern of immune cells in the tumor microenvironment [23], suggesting that CDK1 may affect tumor growth and spread [24] by regulating the immune response. Further bioinformatics analysis revealed multiple key signaling pathways regulated by CDK1, including cell cycle regulation, p53 signaling pathway, and ECM protein and receptor interaction, and the abnormal activation of these pathways was closely related to the development of HCC. Our results not only confirm the multiple biological functions of CDK1 in liver fibrosis and liver CC, but also provide a rationale for CDK1-based targeted therapy. Small-molecule inhibitors against CDK1 have shown antitumor activity in multiple cancer models, providing new hope for the treatment of HCC [25]. When exploring the clinical application of CDK1 as a biomarker for liver fibrosis, we must first recognize its central role in cell cycle regulation, which is not only widely recognized in basic biological studies, but also gradually shows its importance in clinical pathology [26]. The finding that CDK1 expression levels are closely associated with the prognosis of HCC patients has been validated in multiple independent studies, including gene expression profiling [27] in over 500 HCC samples, showing a significant correlation of higher CDK1 expression with worse survival [19]. Further, the expression pattern of CDK1 not only provides potential biomarkers for the early diagnosis of HCC, but also provides new perspectives on the development of personalized treatment strategies. Monitoring of CDK1 mRNA levels by real-time quantitative PCR technology can provide clinicians with real-time information about tumor progression and treatment response [28], thus achieving more precise treatment regimen adjustments. Development and clinical trials of CDK1 inhibitors have also shown promising results [29], especially when used in combination with traditional chemotherapeutic agents, that significantly improve therapeutic efficacy and reduce side effects [30,31]. In the field of immunotherapy, CDK1 expression is closely associated with immune cell infiltration patterns in the tumor microenvironment [32], and this finding provides a rationale for the development of novel therapeutic strategies based on immune checkpoint inhibitors. By analyzing the relationship between CDK1 expression and tumor-infiltrating lymphocytes (TILs) [33], researchers can more accurately predict patients’ responses to immunotherapy, thus achieving more accurate individualized immunotherapy. Despite the great potential of CDK1 as a biomarker, its clinical translation still faces many challenges [34,35,36]. How to standardize the detection method of CDK1 to ensure its consistency and reliability in different laboratories and clinical settings is an urgent problem. The dynamic changes in CDK1 expression and its relationship with tumor heterogeneity also require further studies to clarify the tumor heterogeneity.

CDK1 plays an important role in the activation and proliferation of hepatic stellate cells (HSCs) by regulating liver fibrosis [37]. It was shown that inhibition of CDK1 activity can downregulate the proliferation of HSCs and reduce ECM synthesis, thereby inhibiting the progression of liver fibrosis [7,32,38]. Although the direct application of CDK1 inhibitors in the treatment of liver fibrosis is still under research, some indirect evidence shows that TCM liver recovery can be significantly inhibited by regulating the CDK1-related signaling pathway (such as MAPKAP-1 signaling pathway), which may be related to the indirect regulation of CDK1 [39].

The direct application of CDK1 inhibitors in the treatment of cancer is being gradually explored. Several studies have shown that CDK1 inhibitors were able to significantly inhibit the growth and metastasis of HCC cells and improve the survival rate of patients [29]. However, due to the complexity and heterogeneity of HCC, the therapeutic efficacy of a single CDK1 inhibitor is limited, and it usually needs to be combined with other therapies (for example, immunotherapy, targeted therapy, etc.) to achieve optimal efficacy [40]. Different studies have some differences in the mechanism of CDK1 treatment for liver fibrosis and liver cancer. Some studies have focused on the direct inhibition of CDK1 on HSCs or HCC cell proliferation, while others focus on the regulation of CDK1-related signaling pathways. Moreover, different studies also showed differences in the choice and dosage of CDK1 inhibitors, which may lead to differences in the results of efficacy assessment. Although CDK1 has demonstrated some potential in treating liver fibrosis and liver cancer, some limitations of the current study remain. The complexity of CDK1 in cell cycle regulation makes its single inhibitory effect limited. Moreover, the specificity and safety of CDK1 inhibitors still needs further validation. Meanwhile, the conversion efficiency between preclinical studies and clinical trials needs to be improved to accelerate the clinical application of CDK1 inhibitors in the treatment of liver fibrosis and liver cancer.

For the bioinformatics, survival analysis, and immune infiltration analysis methods used in this study, we first focused on the complexity of bioinformatics and the analysis and its profound impact on the results. Bioinformatics tools, such as the Gene Expression Comprehensive Analysis (GEO) database and Gene Ontology (GO) analysis, provide us with a macro-to-micro perspective, revealing the key role of CDK1 in liver fibrosis and liver cancer. Survival analysis, especially the application of the Kaplan survival curves, provided us with a direct link between CDK1 expression and patient survival [41]. The intuition and statistical rigor of bioinformatics technology are able to provide strong support for clinical decision-making [42]. Immune infiltration analysis, through algorithms, such as CIBERSORT and TIMER, allowed us to gain insight into the role of CDK1 in the tumor microenvironment, especially the influence on immune cell infiltration related to liver fibrosis [43]. This approach is able to provide detailed information about the tumor immune microenvironment for the development of personalized treatment strategies [44,45]. This study mainly relied on gene expression data in public databases, which may not fully reflect the complexity of clinical reality [46]. The immune cell infiltration situation in the tumor microenvironment may be influenced by multiple factors, and our analysis may fail to fully consider the interaction of these factors. Therefore, future studies should combine more clinical data and experimental validation to provide a more comprehensive understanding of the mechanism of CDK1 action in liver fibrosis and HCC.

The potential of CDK1 as a therapeutic biomarker for liver fibrosis and HCC should be further exploited and validated in future studies [47]. The interdisciplinary research approach will combine molecular biology, cell biology, bioinformatics, modern biological technology, and clinical medicine, to deeply analyze the specific mechanism of action of CDK1 in the process of liver fibrosis. For example, by precise editing of the CDK1 gene [48] by CRISPR-Cas 9 technology, researchers can observe direct changes in cell cycle regulation to more accurately assess CDK1 function; combined with high-throughput sequencing technology [49], allowing a comprehensive analysis of other key genes and proteins in the CDK1 regulatory network. These new modern biotechnologies will be even more informative for developing new therapeutic strategies.

In terms of clinical application, the expression level of CDK1 can be used as a basis for personalized treatment of patients with liver fibrosis. Through techniques such as real-time PCR [50], physicians can evaluate the patients, CDK1 expression status before treatment, and then choose the most appropriate drugs and treatment options. For patients with high CDK1 expression, the use of drugs that inhibit cell cycle progression, such as CDK1 inhibitors, may need to be considered to achieve better therapeutic effects. Meanwhile, long-term clinical follow-up studies will help to evaluate the accuracy and reliability of CDK1 as a prognostic marker, which is important for improving the survival rate and quality of life of patients. Research combined with immunotherapy will also be an important direction in the future. CDK1 not only plays a role in cell cycle regulation, but may also influence immune cell infiltration and activity in the tumor microenvironment.

The use of bioinformatics to study CDK1 as a therapeutic target for liver fibrosis and hepatocellular carcinoma has important potential, but there are many limitations. The histological data of hepatocellular carcinoma and liver fibrosis in public databases (e.g., TCGA, GEO) may come from different populations, different pathological stages, or different assay platforms, leading to batch effects or confounding factors interfering with the results of the analyses. And some datasets lack detailed clinicopathological parameters (e.g., treatment history, cirrhosis background, and viral hepatitis infection status), which limits the precise resolution of the CDK1 mechanism of action. During the multi-stage process of liver fibrosis to hepatocellular carcinoma transformation, the expression and function of CDK1 may change dynamically, and it is difficult for static histological data to reflect such temporal regulation. In addition, CDK1 not only regulates the cell cycle (G2/M phase), but also may cross-talk with other pathways (e.g., p53, Wnt/β-catenin, and EMT) through phosphorylation, and it is difficult for bioinformatics tools (e.g., STRING and KEGG) to comprehensively analyze its dynamic regulatory network. Meanwhile, CDK1 may be involved in non-cell cycle-related functions, such as DNA damage repair and metabolic reprogramming, and traditional enrichment analyses, that may miss these potential mechanisms. In molecular docking, CDK family members (e.g., CDK2/4/6) are highly similar in structure, and small molecules targeting CDK1 may lead to off-target toxicity due to insufficient selectivity, whereas computational simulations (e.g., molecular docking) are difficult to fully predict the actual binding specificity. In liver fibrosis, CDK1 may promote fibrosis through the activation of hepatic stellate cells (HSCs), whereas in hepatocellular carcinoma, it may drive tumor cell proliferation, but existing analyses often ignore cell-type specificity (e.g., without integrating single-cell sequencing data). In terms of animal experiments, bioinformatics-screened CDK1-related targets or inhibitors (e.g., dinaciclib) need to be validated by in vitro/in vivo experiments, but some of the studies are only at the computational prediction stage and lack functional (e.g., knockout, organoid models) or pharmacodynamic validation. And the histological data based on mouse models may have species differences with human diseases, affecting the reliability of target validation.

CDK1, as a key enzyme in cell cycle regulation, shows promising applications in the treatment of liver fibrosis and hepatocellular carcinoma. However, many challenges need to be overcome to realize its clinical applications. Future studies should further develop highly efficient and low-toxicity CDK1 inhibitors and strengthen the interface between preclinical studies and clinical trials to promote the widespread application of CDK1 inhibitors in the treatment of liver fibrosis and hepatocellular carcinoma. It also needs to be combined with multi-omics dynamic analysis. Variable modulator design and precise delivery methods, and the interdisciplinary collaboration of computational biology-experimental oncology-clinical medicine to promote the development of CDK1 targets from mechanism research to clinical translation.

## 4. Materials and Method

In this study, we integrated relevant methods, including differential gene analysis, protein interaction analysis, enrichment analysis, survival analysis, immunohistology-related analysis, and molecular docking simulation, combined with multiple public databases and RNA-seq data, to predict new targets for the treatment of hepatic fibrosis, as well as to study the relevant mechanism of action of the new targets, predict the relevant active ingredients for the treatment, and validate the results by combining them with molecular docking simulations (Figure 10).

### 4.1. Acquisition of RNA-Seq Data of Activated Hepatic Stellate Cells (HSCs)

To investigate differential gene expression in growing and senescent-activated hepatic stellate cells (HSCs), GSE11954 gene expression profiles were downloaded from the Gene Expression Omnibus (GEO, https://www.ncbi.nlm.nih.gov/geo, accessed on 10 August 2024) database.

### 4.2. The Differential Gene Expression Analysis of the Microarray Data

GEO2R (https://www.ncbi.nlm.nih.gov/geo/geo2r/, accessed on 10 August 2024) is a tool for further differential analysis of microarray data in the GEO database, which we can use to compare two or more groups of samples from the GEO dataset to obtain differentially expressed genes. Analysis of differentially expressed genes (DEGs) of human-activated liver stellate cells (HSCs) using GEO2R was performed. The targets obtained from the analysis were standardized, and the adjusted *p*-values < 0.05, |log2FC|> 2 were selected for DEGs.

### 4.3. Protein Interaction (PPI) Networks of DEGs

The STRING database (https://string-db.org/, accessed on 11 August 2024) is a database that searches for possible potential interactions between encoded proteins and builds protein interaction networks. DEGs were submitted to the STRING database, the species was set as “Homo sapiens”, the minimum interaction threshold was set to “highest confidence > 0.4”, and the disconnected free target site was cleared to obtain the PPI protein interaction network. Then, the maximum group centrality (MCC) algorithm of Cytoscape3.8.0 and Cyto NCA (2024.8.11) plug-in can simultaneously set “Degree value” and “Betweenness value” as benchmark parameters to conduct topological analysis of the PPI network and screen key DEGs. Finally, we selected the first genes with Degree and Betweenness values in the PPI network and identified them as the key genes of aging human-activated hepatic stellate cells (HSCs) to inhibit liver fibrosis, that is, in the treatment of liver fibrosis disease.

### 4.4. Enrichment Analysis and Functional Annotation of the 2.4 DEGs

The obtained DEGs were imported into the DAVID (https://davidbioinformatics.nih.gov/, accessed on 12 August 2024) database to obtain relevant biological information. The gene ontology (GO) enrichment analysis and the Kyoto Gene-Genome Encyclopedia (KEGG) enrichment analysis were performed after the screening of the obtained biological information. Through bioinformatics analysis, we obtained information about the biological process of DEGs (biological process, BP), the cellular location of DEGs products (cellular component, CC), the activity of DEGs products at the molecular level (molecular function, MF) and the action pathway of DEGs, and visualized the analysis results.

### 4.5. Analysis of Survival Prognosis Related to CDK1

In order to study the prognostic relationship between the CDK1 gene and liver fibrosis, we used the TCGA database (https://portal.gdc.cancer.gov, accessed on 13 August 2024) and GEO database (https://www.ncbi.nlm.nih.gov/geo/, accessed on 13 August 2024), combined with Kaplan–Meier plotter (http://kmplot.com/analysis/, accessed on 14 August 2024), to analyze the key DEG data and draw the survival curve.

### 4.6. Analysis of Immune Cell Infiltration

The relationship between CDK1 gene expression in Homa cells and each immune cell was determined by the Web server from the TIMER database (https://cistrome.shinyapps.io/timer/, accessed on 16 August 2024). A variety of cells associated with tumor immunity were evaluated for immune infiltration, were analyzed for their immunomodulatory effects, and results were visualized. The GEO database was used to obtain various data sets on hepatocellular carcinoma and make a visual heat map to analyze the expression levels of target genes in different cell types in different data sets in hepatocellular carcinoma.

### 4.7. Analysis of the Interaction of CDK1 with Immunosuppressants and MHC Molecules

Sequence, expression profiles, and interaction data of CDK1 and its potential chemokines and receptors were downloaded from public databases, including NCBI (National Center for Biotechnology Information), Ensembl, and UniProt, to analyze the expression of CDK1 chemokines and receptors. The raw data were cleaned and analyzed by the *t*-test and negative binomial distribution analysis (DESeq2, edgeR) to study the application of CDK1 and immune regulators (immunosuppressants, MHC molecules) in human cancer, and the results were visualized as cluster heatmaps.

### 4.8. Association Between CDK1 Expression and Stage Grade of Hepatocellular Carcinoma in Human

Gene expression data from patients with hepatocellular carcinoma (HCC) were obtained from the TCGA database, while other relevant RNA high-throughput sequencing data were downloaded from the GEO database to validate the discovery of the TCGA data. The expression differences in CDK1 gene in different pathological stages and tumor grades were analyzed using the UALCAN platform, and the results were visualized.

### 4.9. Selection of Receptors and Drugs for CDK1 Action in Human Hepatocellular Carcinoma

The rapid search of the CDK1 gene from the TISIDB database yielded the key receptors for CDK1 action on HCC and the active components for the treatment of HCC.

### 4.10. Molecular Docking and Kinetic Simulation for the Validation of the Effective Components for the Treatment of Hepatocellular Carcinoma

In the PubChem database (pubchem.ncbi.nlm.nih.gov, accessed on 22 September 2024) and the PDB database (www.rcsb.org, accessed on 22 September 2024), the 3D structure of all the active components and the protein crystal structure of CDK1 were obtained, and the above components and targets were matched one-to-one by molecular docking analysis on the CB-Dock 2 platform (http://cadd.labshare.cn, accessed on 22 September 2024).

## 5. Conclusions

In conclusion, in this study, we deeply explored the use of the expression pattern, biological function, and its possibility as a potential biomarker of CDK1 in liver fibrosis and liver CC. The combination of bioinformatics analysis, survival analysis and immune infiltration analysis has revealed the key role of CDK1 in liver fibrosis and liver cancer. The expression level of CDK1 may not only serve as an independent predictor of HCC prognosis, but may also provide new ideas for molecular typing and personalized treatment of HCC. This finding not only enriches our understanding of the molecular mechanisms of HCC, but also provides new perspectives for future clinical applications.

Further, this study not only deepens our understanding of the role of CDK1 in liver fibrosis HCC at the theoretical level, but also provides new tools and strategies for the precise treatment and prognosis evaluation of HCC at the practical level. The transformation and application of these research results are expected to improve the treatment effect and quality of life of liver cancer patients in the future and have important scientific value and clinical significance.

## Figures and Tables

**Figure 1 ijms-26-03816-f001:**
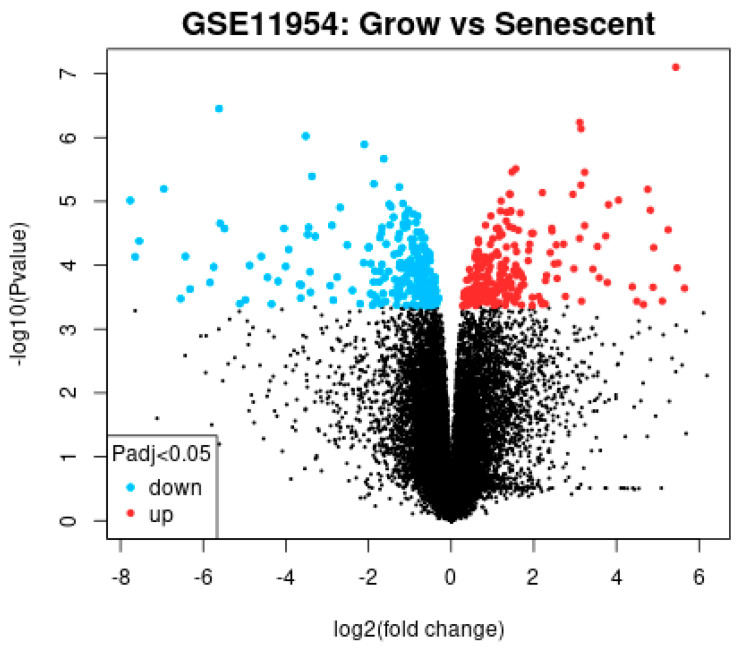
Volcano plot of the differential gene analysis.

**Figure 2 ijms-26-03816-f002:**
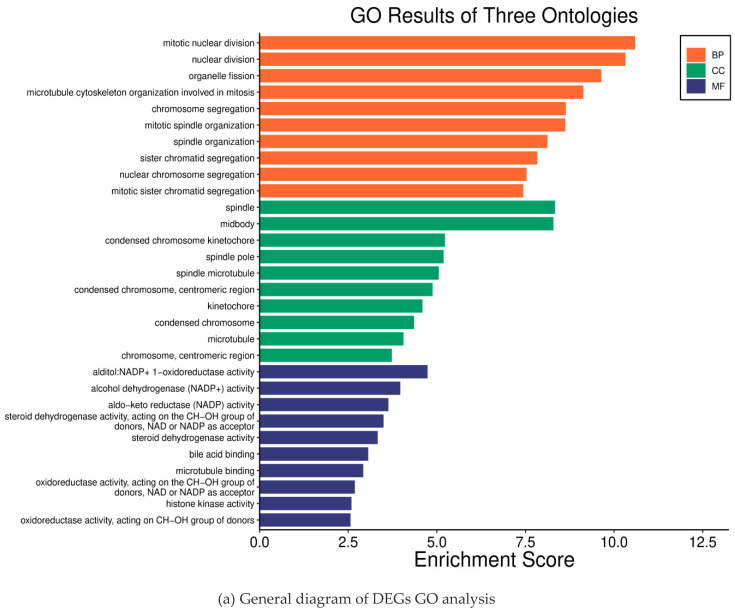
Results of the differential gene enrichment analysis.

**Figure 3 ijms-26-03816-f003:**
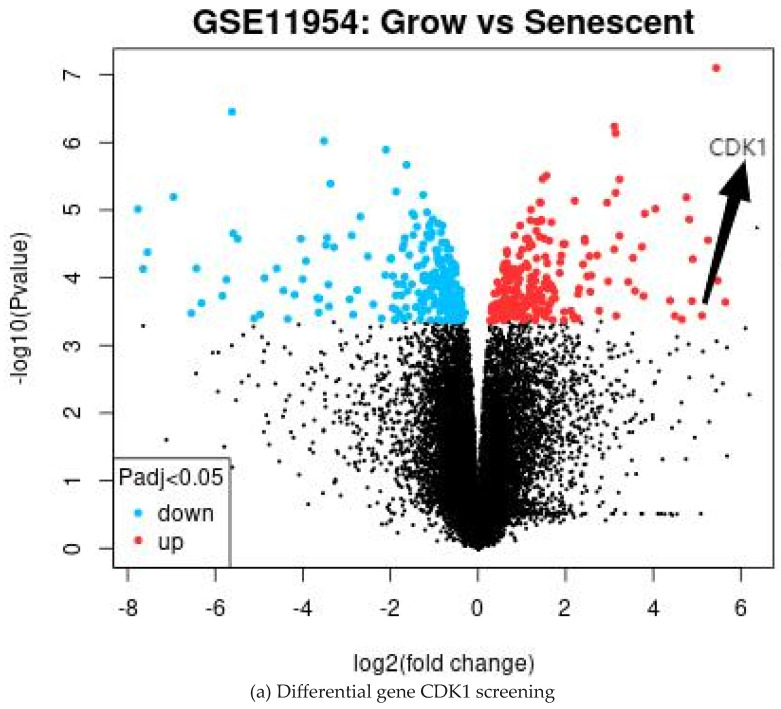
Protein interaction and screening of key differential genes.

**Figure 4 ijms-26-03816-f004:**
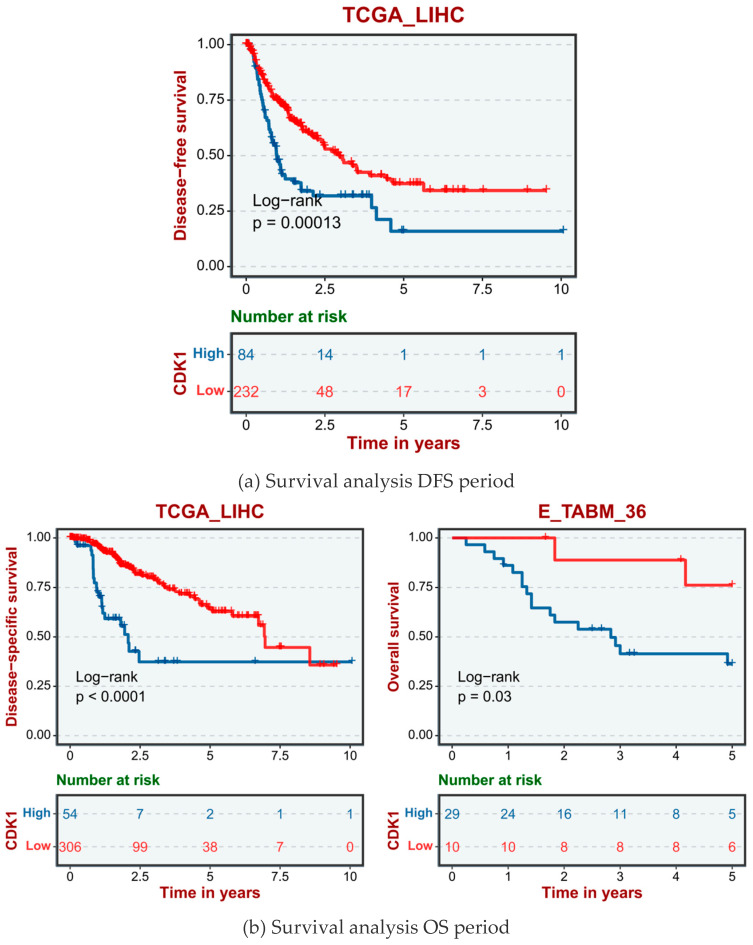
Survival analysis of CDK1 gene expression in normal hepatocytes.

**Figure 5 ijms-26-03816-f005:**
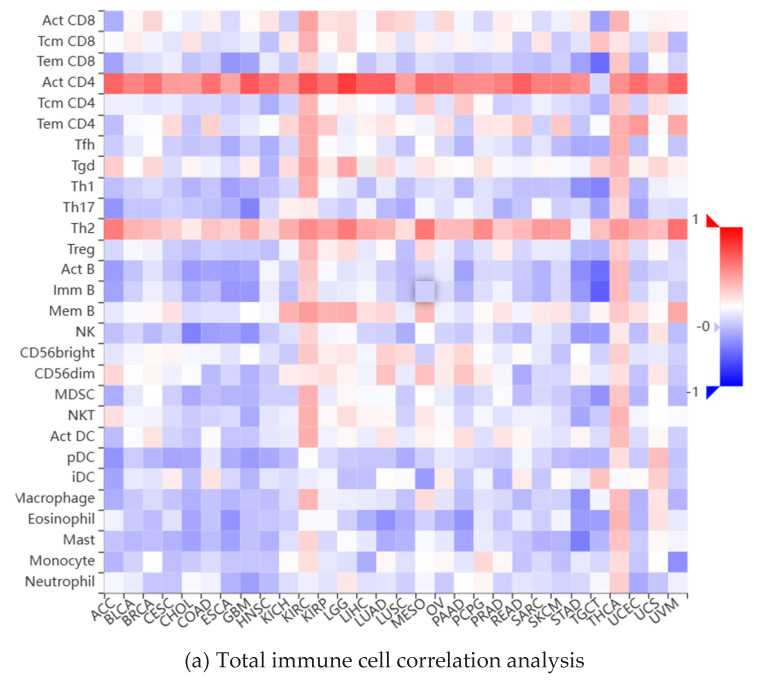
Immune infiltration of the CDK1 gene.

**Figure 6 ijms-26-03816-f006:**
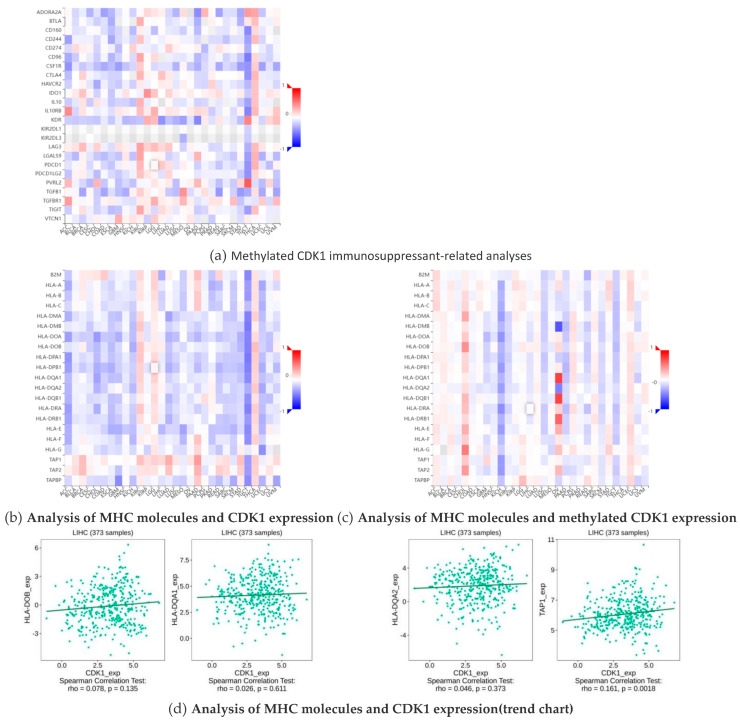
Interactions of CDK1 and immunosuppressants, MHC molecules.

**Figure 7 ijms-26-03816-f007:**
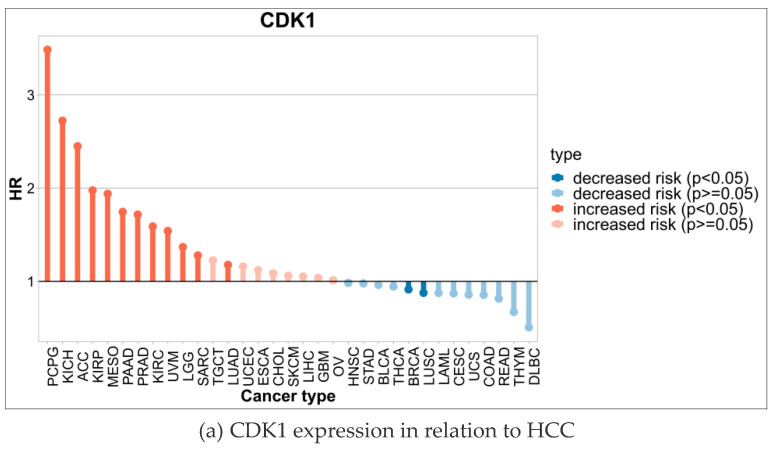
CDK1 expression and tumor grade and staging.

**Figure 8 ijms-26-03816-f008:**
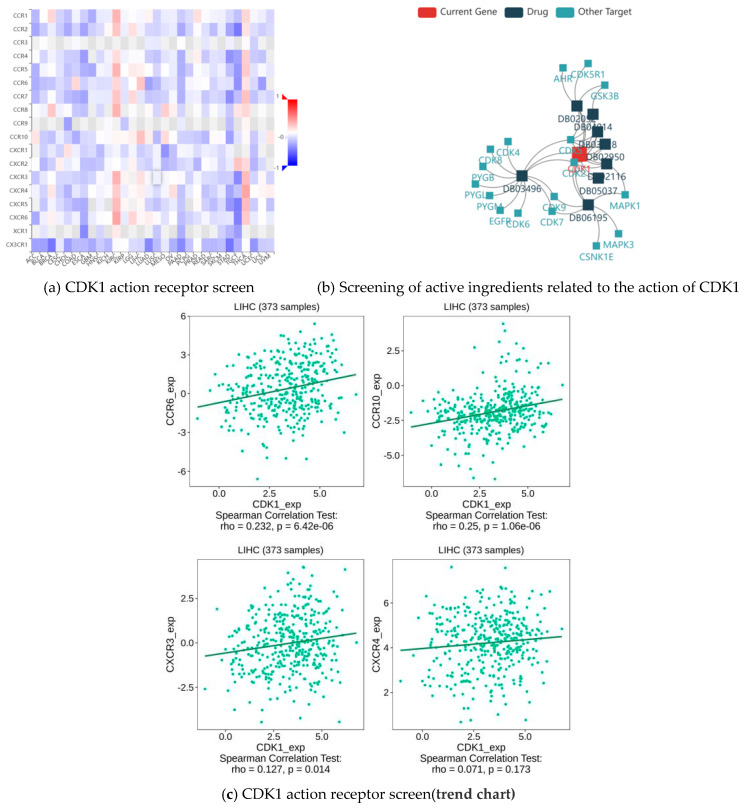
CDK1 action receptors and associated effective therapeutic components.

**Figure 9 ijms-26-03816-f009:**
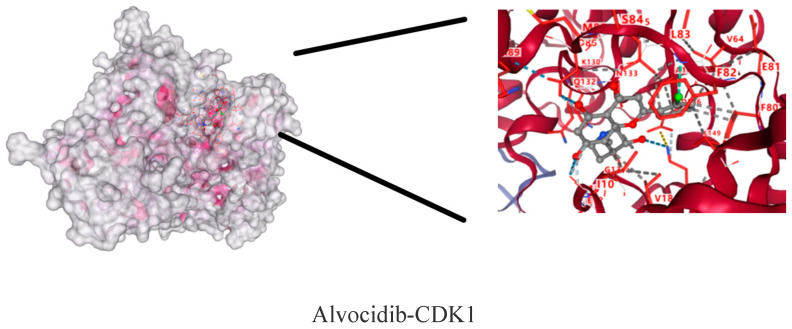
CDK1—active component molecular docking.

**Figure 10 ijms-26-03816-f010:**
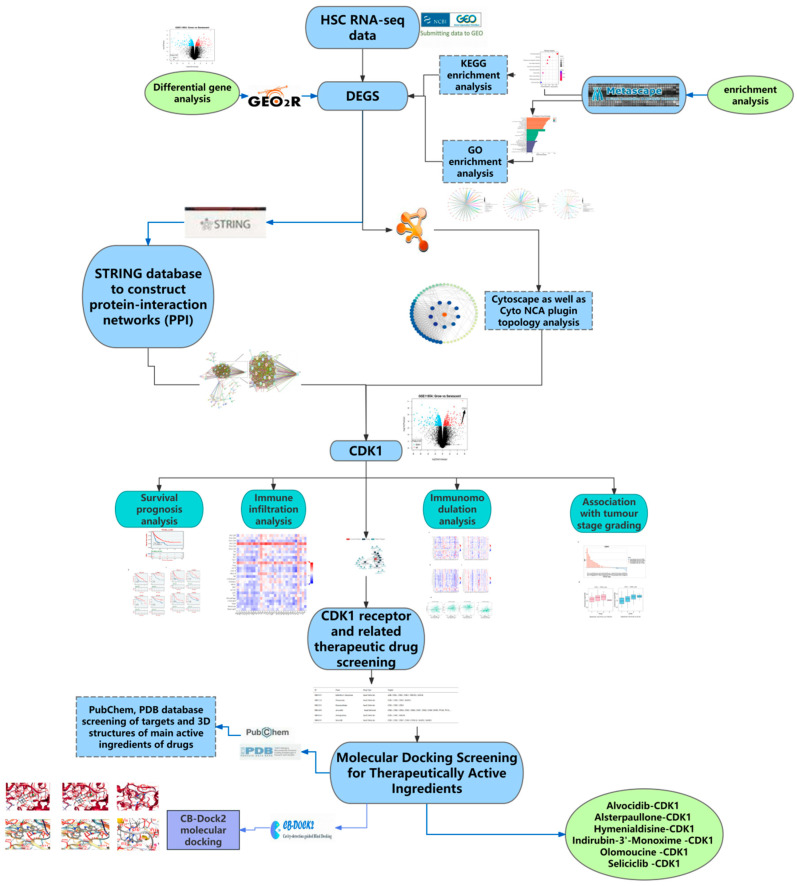
Research on methods and materials.

**Table 1 ijms-26-03816-t001:** Top100 significantly differential genes.

Gene Symbol	adj.*p*.Val	*p* Value	t	B	logFC
TOP2A	0.12789	0.00535	7.3	−1.89819	6.1906911
CALD1	0.05537	0.000559	15.8	0.85216	6.10536365
RRM2	0.38828	0.0432	3.38	−4.55825	5.6877799
CDK1	0.06921	0.00108	12.7	0.08781	5.681935
ANLN	0.04192	0.000229	21.3	1.81079	5.6470256
DLGAP5	0.1093	0.00366	8.33	−1.4181	5.58844865
NEK2	0.03218	0.000111	27.2	2.50046	5.4713496
CDKN3	0.06542	0.000872	13.6	0.33851	5.4477173
PRC1	0.00433	7.92 × 10^−8^	307	4.6163	5.438704
HMMR	0.11977	0.00466	7.66	−1.72192	5.4364495
TOP2A	0.10102	0.00287	9.06	−1.11079	5.34630875
APOBEC3B	0.1972	0.0135	5.26	−3.07882	5.2771654
ASPM	0.02438	0.0000279	43.2	3.50872	5.25215545
SHROOM3	0.07134	0.00123	12.1	−0.07062	5.1339738
CDK1	0.0469	0.000364	18.3	1.32791	5.11074285
MIR503///MIR424///MIR503HG	0.26429	0.0231	4.31	−3.76446	4.9491692
CDKN3	0.02826	0.0000529	34.9	3.0913	4.9007493
CDK1	0.04122	0.000222	21.6	1.84474	4.88621195
MYH10	0.10263	0.003	8.92	−1.16686	4.8265022
DEPDC1	0.02348	0.0000137	54.8	3.86617	4.82202725
NUF2	0.06714	0.00096	13.2	0.22503	4.80625185
CDC20	0.02141	0.00000649	70.4	4.13784	4.7577602
SHCBP1	0.41379	0.048	3.24	−4.69101	4.746537
NUSAP1	0.04944	0.000412	17.5	1.19152	4.6542621
MEGF6	0.14361	0.00702	6.64	−2.24408	4.6460274
KIAA0101	0.19579	0.0133	5.29	−3.05921	4.59005025
CCNB2	0.06324	0.000747	14.3	0.51913	4.54231145
TK1	0.07043	0.00116	12.4	0.00481	4.53425635
ST6GALNAC5	0.04696	0.000367	18.2	1.31903	4.4934988
TRIP13	0.21995	0.0166	4.87	−3.34373	4.42963645
KIF11	0.04077	0.000217	21.7	1.86588	4.3848553
ANLN	0.07284	0.00134	11.8	−0.17191	4.37412475
UBE2C	0.09333	0.0024	9.63	−0.88967	4.129496
BIRC5	0.02343	0.00000957	61.8	4.00897	4.04899235
GALNT15	0.107	0.00348	8.48	−1.35462	4.0395688
PBK	0.10906	0.00364	8.35	−1.40896	4.0233828
CCNB1	0.18706	0.0121	5.46	−2.94424	3.98987825
SYNPO2	0.17546	0.0107	5.71	−2.78445	3.9871827
TK1	0.08096	0.00175	10.7	−0.49598	3.942686
ID4	0.19244	0.0128	5.36	−3.01187	3.8204956
CEP55	0.02348	0.0000113	58.5	3.94689	3.8085747
CENPH	0.03812	0.000187	22.8	2.01352	3.78016685
KNL1	0.02538	0.0000349	40.1	3.37228	3.74212255
COL4A2	0.13932	0.00662	6.78	−2.16919	3.7290946
ITGA11	0.20015	0.014	5.19	−3.12318	3.72201025
PTN	0.08079	0.00173	10.8	−0.4872	3.6569635
SORBS1	0.17335	0.0105	5.76	−2.75316	3.60330895
OXTR	0.03626	0.000157	24.2	2.18018	3.58367625
KIF20A	0.12046	0.00473	7.62	−1.74108	3.54996585
NREP	0.186	0.012	5.48	−2.92871	3.5390412
GPAT3	0.14374	0.00703	−6.63	−2.24599	−4.0068785
LRRN3	0.02391	0.0000265	−43.9	3.53655	−4.0428889
GDF15	0.03792	0.000179	−23.2	2.05611	−4.1874975
LOC101930400///AKR1C2///AKR1C1	0.26356	0.023	−4.32	−3.75854	−4.2147407
H2BFS	0.23672	0.0191	−4.63	−3.52323	−4.2734285
HIST1H4H	0.0728	0.00132	−11.8	−0.1598	−4.2883313
KCNJ2	0.12826	0.00549	−7.23	−1.93174	−4.3042694
F2RL1	0.04937	0.000407	−17.6	1.20609	−4.3446080
MMP10	0.1383	0.00651	−6.82	−2.1471	−4.3788025
CD24	0.16218	0.00915	−6.05	−2.58109	−4.4126261
DUSP6	0.06921	0.00107	−12.7	0.09775	−4.4189292
MDM2	0.03616	0.000155	−24.3	2.19471	−4.4415299
CNIH3	0.17998	0.0113	−5.61	−2.84847	−4.4545942
HIST1H2BC///HIST1H2BI///HIST1H2BE///HIST1H2BF///HIST1H2BG	0.03057	0.0000729	−31.3	2.84933	−4.5961153
HIST1H1C	0.30462	0.0292	−3.94	−4.06465	−4.7696097
CORO2B	0.19916	0.0138	−5.21	−3.11184	−4.7864721
HIST1H2BC	0.06714	0.000945	−13.2	0.24434	−4.7936413
HIST1H4H	0.17579	0.0108	−5.7	−2.79027	−4.8343656
MIR146A	0.03218	0.000101	−28.1	2.58075	−4.8751408
PDK4	0.07133	0.00121	−12.2	−0.05298	−4.8824006
CTSS	0.23602	0.019	−4.64	−3.51716	−4.8909467
PLAU	0.08098	0.00175	−10.7	−0.49883	−4.9086587
RRAD	0.06324	0.000748	−14.3	0.51759	−4.9342201
GK	0.04656	0.00035	−18.5	1.37074	−4.9782953
BTBD11	0.11079	0.0039	−8.15	−1.49722	−5.0217412
CLU	0.04927	0.000402	−17.7	1.22063	−5.1144443
AKR1B10	0.05446	0.00053	−16.1	0.91266	−5.1255345
CD24	0.09974	0.0028	−9.14	−1.08137	−5.2352129
DCBLD2	0.06186	0.000702	−14.6	0.59178	−5.3608140
LINC00973	0.02391	0.0000267	−43.9	3.53382	−5.4841241
MMP1	0.13772	0.00646	−6.84	−2.13708	−5.5256527
AADAC	0.02391	0.0000221	−46.7	3.63589	−5.5888679
SERPIND1	0.00966	0.00000035	−186	4.56539	−5.6156946
KYNU	0.06773	0.001	−13	0.1743	−5.6242355
ST3GAL6	0.03218	0.000107	−27.6	2.5333	−5.7433365
PLAU	0.31948	0.0315	−3.83	−4.16154	−5.7927380
SLC16A6	0.03812	0.000186	−22.9	2.01797	−5.8349129
CD24	0.07227	0.00127	−12	−0.11069	−5.9268551
THBD	0.12169	0.00481	−7.58	−1.76274	−5.9403651
CD24	0.07242	0.00128	−11.9	−0.12211	−6.0572334
IL13RA2	0.04214	0.000237	−21.1	1.77762	−6.3166654
CXCL5	0.03057	0.0000727	−31.3	2.85071	−6.428763
SPP1	0.09665	0.0026	−9.37	−0.99003	−6.4387306
HSD11B1	0.04578	0.000335	−18.8	1.41684	−6.5452208
NEFL	0.02141	0.00000638	−70.8	4.14284	−6.9522091
MMP3	0.27605	0.025	−4.18	−3.86739	−7.1158428
THBD	0.02666	0.0000419	−37.7	3.25337	−7.5474292
IL24	0.05373	0.000515	−16.2	0.94497	−7.646795
RAB27B	0.03057	0.0000737	−31.2	2.84013	−7.6470385
EHF	0.02343	0.00000969	−61.5	4.00437	−7.7647490

**Table 2 ijms-26-03816-t002:** Related parameters of the PPI network.

Parameter	Numerical Value
**number of nodes:**	**85**
**number of edges:**	**447**
**average node degree:**	**10.5**
**expected number of edges:**	**82**
**PPI enrichment *p*-value:**	**<1.0 × 10^−16^**

**Table 3 ijms-26-03816-t003:** Active components for treating HCC.

ID	Name	Drug Type	Targets
DB02052	Indirubin-3′-Monoxime	Small Molecule	AHR, CDK1, CDK2, CDK5, CDK5R1, GSK3B
DB02116	Olomoucine	Small Molecule	CDK1, CDK2, CDK5, MAPK1
DB02950	Hymenialdisine	Small Molecule	CDK1, CDK2, CDK5
DB03496	Alvocidib	Small Molecule	CDK1, CDK2, CDK4, CDK5, CDK6, CDK7, CDK8, CDK9, EGFR, PYGB, PYGL,…
DB04014	Alsterpaullone	Small Molecule	CDK1, CDK5, GSK3B
DB06195	Seliciclib	Small Molecule	CDK1, CDK2, CDK7, CDK9, CSNK1E, MAPK1, MAPK3

**Table 4 ijms-26-03816-t004:** Docking and binding energy of CDK1-active component molecules.

Active Ingredients	Binding Afinity (eV)
	CDK1
Indirubin-3′-Monoxime	**−2.6**
Olomoucine	**−6.7**
Hymenialdisine	**−6.3**
Alvocidib	**−8.7**
Alsterpaullone	**−7.8**
Seliciclib	**−8.0**

## Data Availability

All datasets, analytical scripts, and computational workflows generated or analyzed during this study are publicly available in the GEO, TIMER, TISIDB and TCGA databases. Detailed descriptions of tools and parameters ensure full reproducibility of the results.

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
