# Peer review of "Identification of CDK1 as a Biomarker for the Treatment of Liver Fibrosis and Hepatocellular Carcinoma Through Bioinformatics Analysis"

_ijms, 2025, doi:10.3390/ijms26083816_

Round 1

Reviewer 1 Report

Comments and Suggestions for Authors

minor revision. It is clear that CDK1 is a very important factor is a single biomarker, but when used jointly with other biomarkers, it is an important factor as a diagnostic marker and therapeutic target for liver cancer with fibrosis, which is considered an important paper.

Author Response

minor revision. It is clear that CDK1 is a very important factor is a single biomarker, but when used jointly with other biomarkers, it is an important factor as a diagnostic marker and therapeutic target for liver cancer with fibrosis, which is considered an important paper.[Thank you for your suggestions, and thank you for the affirmation that I have made the exact changes based on the reviewers' comments!]

Reviewer 2 Report

Comments and Suggestions for Authors

The paper “Identification of CDK1 as a Biomarker for the Treatment of  Liver Fibrosis and Hepatocellular Carcinoma Through Bioinformatics Analysis” applies advanced statistics to work through the data from different database in order to determine a biomarker for fibrosis-associated hepatocellular carcinoma. It is an interesting approach that uses all the advances of the bioinformatics analysis for the identification of the most prominent biomarkers in the treatment of liver fibrosis and hepatocellular carcinoma. Exploring new  for new therapeutic strategies is worthy fro publication. Nevertheless, before the paper is considered for publication some points should be improved and some comments needs to be addressed.

  1. Line 36 – please do not start a paper with “today”.
  2. Line 45 “Liver fibrosis is a necessary stage for the development of many chronic liver diseases to cirrhosis…” – the word “necessary” is inappropriate here. “Liver fibrosis is a first step in the development of many chronic…”
  3. 2. Materials Method“ should be Materials and Method or Methodology
  4. The Figure after the title Materials and Methods needs to be given caption, numeration and to be mentioned somewhere in the text before it appears...
  5. The title of the paper suggest that CDK1 is a biomarker for treatment and as such is used throughout the paper. But, it is maybe better to address to CDK1 as biomarker for treatment response
  6. Before conclusion add limitations of this study.

7.Was there any bias? How have you eliminated the possible bias? what have you applied to eliminate the false positive and the false negative cases? What was the sensitivity and precision of  the study? How have you measured that?

Author Response

1、第 36 行 – 请不要以 “today” 开头。[感谢您的更正,我们同意该评论,因此我们将第 36 行的“今天”更改为“最近”。

2、第 45 行“肝纤维化是许多慢性肝病发展为肝硬化的必要阶段......”——“必要”一词在这里不合适。肝纤维化是许多慢性肝病发展的第一步......“[感谢您的更正,我们同意该评论,因此我们将第 45 行的原始句子修改为”肝纤维化是许多慢性肝病进展为肝硬化的第一阶段。’]

3、“2.Materials Method“应为 Materials and Method 或 Methodology。[感谢您的更正,我们同意您的评论,因此我们将第 104 行修改为“材料和方法”。

4、标题 Materials and Methods 后面的 Figure 需要给出标题、编号并在正文中出现之前在正文中的某处提及......[感谢您的更正,我们同意此评论,因此我们在第 106 行插入“图 1 方法和材料研究”以在标题后解释图片,并在第 104 行后插入描述,以便提及图 1。

5、论文的标题表明 CDK1 是一种治疗生物标志物,因此在整篇论文中都有使用。但是,将 CDK1 作为治疗反应的生物标志物可能更好。[感谢您的更正,我们同意此评论,但本文希望澄清 CDK1 用作对肝脏相关疾病治疗反应的生物标志物,并且不想研究明确的通路,而是想从宏观角度研究 CDK1 靶标。

6、在结论之前增加本研究的局限性。[感谢您的更正,我们同意此评论,因此我们在第 474 行之后讨论此研究的局限性。

  1. 有什么偏见吗?您是如何消除可能的偏见的?您应用了什么来消除假阳性和假阴性情况?这项研究的敏感性和精确度如何?您是如何衡量的?[感谢您的更正,我们同意此评论。在本研究中,我们整合了来自不同数据库来源(如 TCGA、GEO、ICGC 等)的肝细胞癌和肝纤维化数据,并使用 ComBat 算法消除了批量效应。CDK1 表达的一致性也通过按病因 (病毒/非病毒) 、病理分期 (Child-Pugh 分类) 和分子分期 (例如增殖性肝细胞癌) 进行分组来验证。还使用 GSEA 和 GSVA 进行通路富集,以比较结果的重叠。在分子对接中,使用 CB-dock2 与 AutoDock 交叉验证分子对接,结合能筛选后(例如 ΔG < -7 kcal/mol),通过 ADMET 预测排除不良的药物样化合物(例如 SwissADME),并使用灵活对接(例如 Induced Fit Docking)来捕获构象变化并减少由于刚性对接引起的假阴性化合物的数量。在差异表达分析中使用 Benjamini-Hochberg 校正 (FDR < 0.05),并将 |log2FC|>1 设置为显著性阈值。在敏感性方面,通过 Kaplan-Meier 曲线分析 CDK1 表达对肝细胞癌患者生存预后的预测能力。为了精确起见,通过 GO 或 KEGG 富集的 p 值校正分析与 CDK1 直接相关的重要通路的比例,并得到文献支持的验证。在预测的结合化合物中,通过分子结合能的目视验证来确认有效化合物的比例。

Round 2

Reviewer 2 Report

Comments and Suggestions for Authors

The authors have answered to all of my questions. It is upon the Editor to decide whether the paper is worthy of publication.